# Analysis of Motives and Factors Connected to Suicidal Behavior in Patients Hospitalized in a Psychiatric Department

**DOI:** 10.3390/ijerph19106283

**Published:** 2022-05-22

**Authors:** Aneta Bednarova, Natasa Hlavacova, Jan Pecenak

**Affiliations:** 12nd Department of Psychiatry, Faculty of Medicine, University Hospital of L. Pasteur Kosice, Pavol Jozef Safarik University, 04011 Kosice, Slovakia; aneta.bednarova@upjs.sk; 2Institute of Experimental Endocrinology, Biomedical Research Center, Slovak Academy of Sciences, 84505 Bratislava, Slovakia; 3Department of Psychiatry, Faculty of Medicine, University Hospital Bratislava, Comenius University, 81369 Bratislava, Slovakia; janpecenak@gmail.com

**Keywords:** suicide, self-injury, motivation, predictors, Columbia Suicide Severity Rating Scale, qualitative analysis

## Abstract

(1) Background: This study aimed to investigate the motives and factors connected to suicidal behavior in 121 hospitalized patients with intentional self-harm (diagnosis X 60-81 according to the ICD-10); (2) Methods: Suicidal behavior of the patient was assessed from data obtained by psychiatric examinations and by the Columbia Suicide Severity Rating Scale. Analysis of data to identify the patients’ reason and motives behind suicidal behavior in a group of patients with a suicide attempt (SA, *n* = 80) and patients with Non-Suicidal Self-Injurious Behavior (NSSIB, *n* = 41) was carried out; (3) Results: Results showed that patients with affective disorder have a 19-times higher rate of SA against other diagnoses. Patients with personality disorders have a 32-times higher rate of NSSIB than patients with other diagnoses. Living alone and the absence of social support increased the likelihood of SA. Qualitative data analysis of patients’ statements showed different themes in the justification of motives for suicidal behavior between SA and NSSIB cases. Significant differences were shown for non-communicated reasons, loneliness, social problems, extortion, and distress; (4) Conclusions: The evaluation of patients’ verbal statements by qualitative analysis during the psychiatric examination should be considered in clinical practice. It should be considered to include self-poisoning in the criteria of the Non-suicidal Self-Injury diagnostic categories.

## 1. Introduction

About a million people a year die from suicide, which equates to one person every 40 s. It is the fourteenth most common cause of death worldwide. On average, there were 11 deaths by suicide per 100,000 population across EU countries in 2017 [1]. From a clinical perspective, it is particularly important to identify the nature and predictors of suicidal behavior. Although several variables have been associated with suicidality, their usefulness in predicting future suicide remains unresolved [2]. Management of suicidality calls for a comprehensive approach to assessment. Assessment should focus on past suicidal behavior, openly addressing ongoing suicidal ideas, and psychosocial needs [3]. Suicidal behavior shows marked differences between genders, age groups, geographic regions, and socio-political realities, and is variably associated with different risk factors, underscoring a likely etiological heterogeneity. Although there is no effective algorithm to predict suicide in clinical practice, improved recognition and understanding of clinical, psychological, sociological, and biological factors may facilitate the detection of high-risk individuals [4].

The research on suicide and suicidal behavior is much more complicated by inconsistent terminology over the years [5]. Suicide is defined as death caused by self-directed injurious behavior with an intent to die, suicide attempt is defined as a nonfatal, self-directed, potentially injurious behavior with an intent to die, suicidal ideation is defined as thinking about, considering, or planning suicide [5,6]. The Kreitman’s term of parasuicide was used to label all of the non-accidental, self-poisoning, or self-injury that did not result in death, regardless of the intention of the act, and the term non-suicidal self-injury (NSSI) arose from this concept. It may differ from a suicide attempt with respect to various factors, such as intent, lethality, chronicity, methods, cognitions, reactions, demographics, and prevalence [7]. Much of the literature on NSSI has focused on young people. Comparatively few studies were carried out on adults [8].

There is an unmet need to reduce suicide deaths, and to reduce the pain and suffering arising from nonlethal suicide attempts. One approach to reducing suicide attempts and deaths is by clarifying what motivates the subject to commit suicide. Given that NSSI significantly increases the risk of future suicide [9], and suicide attempts are associated with a worse treatment course and increased risk of mortality, it is important to understand the motives and reasons why certain individuals engage in NSSI, whereas others engage in suicide attempts [10]. Motivations and reasons for suicide have rarely been investigated at a qualitative level [11].

Over the past few years, the relationship between suicidal behavior and mental disorders was the focus of several studies or meta-analyses and has generated important debate [12]. Suicidal thoughts and behavior are more common than suicide and predict future suicide attempts [13,14]. The etiology of suicidal ideation and behavior is multi-factorial, although one of the most common risk factors is having a psychiatric disorder [15,16]. Several psychological autopsy studies have supported high rates of psychiatric disorders among individuals who die by suicide [17,18]. Arsenault-Lapierre et al. [19] reported in their meta-analysis that 87.3% of suicide completers were diagnosed with a psychiatric disorder prior to the suicide. Existing reviews indicate that major depressive disorder, dysthymia, anxiety disorders, bipolar disorder, and schizophrenia are significantly associated with suicide risk [12,20,21]. Although psychopathology or the presence of a mental disorder mediate suicide risk in a substantial proportion, other factors, such as gender and geographical differences, also play a significant role [19]. Importantly, social isolation and physical distancing were found to be serious risk factors for suicidal behavior during the COVID-19 pandemic [22].

Suicide is a serious problem with a complex background in terms of motivation and execution. People who have attempted to die by suicide represent an important segment of the patients admitted to psychiatric departments. The issue is much more complicated by the fact that patients are admitted to the emergency departments without further specification of a suicide attempt, which distorts the clinical evaluation and statistics. New concepts (e.g., NSSI) are emerging for the classification of suicide attempts, but self-poisoning, which is a common reason for hospitalization, is overlooked. The evaluation of the background of NSSI and suicide attempts, their relation to existing mental disorders, prediction of a repeated suicide attempt, and correctly selected treatment of primary mental disorders requires complex diagnostics and therapeutic approaches. The aim of this study was to analyze the motives and factors connected to suicidal behavior in patients hospitalized with a diagnosis X60 to X81 “Intentional self-harm”, according to the ICD-10 as these diagnostic codes and their descriptions do not allow more specified assessment of suicidal behavior. After clinical assessment and administration of the Columbia Suicide Severity Rating Scale, patients were divided into two groups: the group of patients with suicide attempt (SA) and the group of patients with Non-Suicidal Self-Injurious Behavior (NSSIB). These groups were compared according to demographic and clinical characteristics, and a qualitative analysis of statements about the reasons for suicidal behavior was performed to find the difference between the groups.

## 2. Materials and Methods

### 2.1. Subjects

This study included 121 consecutive patients older than 15 years admitted to the psychiatric department with a diagnosis of X60–X81 according to the ICD-10 listed among the reasons for hospitalization. Patients were examined within 5 days from their admission to the department. The duration of the interviews, including the administration of the Columbia Suicide Severity Rating Scale, was approximately one hour. Patients with severe psychotic symptoms, delirium, and severe cognitive dysfunctions were excluded. The study was performed at the University Hospital of L. Pasteur Kosice, Slovakia, during a period of six years. All of the subjects provided informed consent and the protocol was approved by the Ethics Committee of the University Hospital of L. Pasteur. The study was conducted in accordance with the Helsinki Declaration.

### 2.2. Assessments

#### 2.2.1. Socio-Demographic and Clinical Variables

Patients who consented to participate in the study were interviewed by the first author of the study, a certified unbiased clinical psychiatrist, within 5 days of admission. The interview was based on the prepared protocol that included questions about socio-demographics and clinical data, such as sex, age, type of cohabitation, education, accompanying physical illness, previous diagnosis and treatment for mental disorder, current treatments, previous hospitalization, and childhood adversity or trauma. Additional data, mainly information collected at the time of admission, were obtained from the medical documentation available in the department. A routine face-to-face psychiatric examination was performed. The relationship with patients was not established before the start of the investigation.

#### 2.2.2. Suicidal Behavior and Qualitative Data Analysis

In all of the patients, the suicidal behavior of the patient was assessed by the Columbia Suicide Severity Rating Scale (C-SSRS). The C-SSRS is a validated and reliable instrument that measures current and past suicidal ideation, suicide attempts, preparatory behaviors, as well as NSSIB, and deliberate self-harm behaviors performed with no intent to die [23,24,25]. The first author of the study, a psychiatrist (MD, female, unbiased,) responsible for the assessments was systematically trained and certified for the administration of the C-SSRS. The Slovak version of the scale was adopted. For the screening assessment of social support, the item “social supports lacking” from the SAD PERSONS [26] scale was used.

The suicidal behavior of each patient was evaluated using the C-SSRS, according to recommendations. The study sample was divided into sub-groups based on demographic data, psychiatric diagnosis, and suicidal behavior. For the statistical analyses, two main sub-groups of suicidal behavior were identified: NSSIB (*n* = 41) and SA (*n* = 80). The SA group contained the patients who attempted suicide with the intention to die. In the NSSIB group, the intention to die was missing. Both groups included the self-poisoning patients, as we focused on monitoring the incidence of intoxications compared to other methods of suicide.

In order to investigate the patients’ reasons and motives for NSSIB and SA, the transcribed interviews with patients were analyzed using qualitative content analysis with MAXQDA 11 software [27]. This method was used to organize the data into codes (themes) and sub-codes (sub-themes). First, the interview statements were analyzed line by line, and the initial codes were identified, then the codes were sorted into subthemes based on differences and similarities. Finally, the subthemes were grouped into themes, with definitions stated for each theme. Nine qualitatively different themes were identified (see Results).

We used the Consolidated Criteria for Reporting Qualitative Studies (COREQ) checklist to report our data (see Appendix A).

### 2.3. Statistical Analyses

Descriptive statistics to describe groups and tracked markers were used. Pearson’s chi-square (χ^2^) was used for comparisons of frequencies in the categories of the variables studied in the patients who had either engaged in NSSIB or SA. Binary logistic regression analysis was applied to test the connection between the explanatory variables and the dependent variable (SA vs. NSSIB). Binary logistic regression analysis with the step of adding significant variables (stepwise forward method) and decommissioning non-significant variables was used. Stepwise logistic regression allowed, from among the many studied factors influencing the “authenticity” of SA, the creation of a robust predictive model that contained only the relevant predictors. The overall level of statistical significance was defined as *p* < 0.05.

## 3. Results

The study sample included a total of 121 hospitalized patients with diagnoses X60 to X81 (ICD-10) with an average age of 39.3 years. Basic sociodemographic and clinical variables are shown in Table 1. Women represented 44.6% of the sample subjects. Approximately 32% of patients were married at the time of admission, and 50.4% of patients were without children. The largest part of the sample consisted of patients with a diagnosis of mood disorders (30.6%). Nine cases were diagnosed with behavioral disorders, of whom three patients were from the SA group. Four of eight patients with a diagnosis of mental retardation were identified as SA cases. In our sample, the highest number of suicides occurred at the beginning of the week (Monday to Wednesday—more than 60%), and the lowest suicide rate was recorded on Thursday. A total of 58.7% of subjects had been previously diagnosed with a mental disorder. Approximately half of the patients had a positive alcohol test when admitted to the hospital. A total of 41.3% of patients had repeatedly attempted suicide, six cases had a positive indication of suicide in the family history. Stress in childhood and adolescence was reported by 68% of the patients (most often described as conflicted coexistence with parents, fights, and quarrels). Regarding intentional self-harm (according to ICD-10), the most common was intentional self-poisoning (58.7%), followed by intentional self-harm with a sharp object (28.9%) (Table 1).

Based on the character of suicidal behavior assessed by psychiatric examinations and the C-SSRS scale, the patients were divided into the group engaged in SA (*n* = 80) and the group engaged in NSSIB (*n* = 41). Thus, 34% of the study sample were subjects with NSSIB and 66% were patients with SA. In order to find out the relevant predictors of suicidal behavior regarding clinical and socio-demographic variables, binary logistic regression was used. First, a diagnosis of mental disorder was approached. In the analyses, seven groups of psychiatric diagnoses according to the ICD-10 were included (F1x.x, F2x.x, F3x.x. F4x.x, F6x.x, F7x.x, and F9x.x). Regression analysis showed that a diagnosis of affective disorder significantly increases the rate of SA over the other diagnostic groups, by up to 19 times (OR = 19.47, *p* < 0.001). On the contrary, patients with a diagnosis of personality disorder have a 32-times higher rate of NSSIB, compared to other diagnostic groups of the patients (OR = 32.692, *p* < 0.001).

Next, marital status and suicidal behavior were investigated. We predicted the occurrence of a serious suicidal attempt in four groups (single, married, divorced, widowed) at the same time in logistic regression. The “marital status” entered the analysis as a categorical variable and the category “single individuals” was chosen as the reference group. Results clearly showed that SA was identified in fewer cases in married patients compared to singles (OR = 0.26; *p* < 0.01). On the other hand, there was a trend of more suicidal attempts performed in divorced (OR = 1.49). However, the result did not reach statistical significance. The differences between single and divorced, and single and widowed were not significant.

Using a regression analysis, we calculated the ratio of the chances for groups of suicide attempters with a lack of social support compared to those with social support (OR = 4.16, *p* < 0.001). The likelihood of a suicide attempt increased significantly in patients with a lack of social support (four times).

Qualitative analysis of the interview data by MAXQDA software was employed to investigate the patients’ reasons and motives for suicidal behavior. Nine qualitatively different themes were identified (Table 2). A special category of “Non-communicated” represented the coded summaries for cases in which the reason could not be determined (for example, most of these were cases of severe depression). The other themes that emerged from the analysis were: “Impulsive”—Emphasizes the fact that the attempt has been unexpected and exceptional for the patient; “Extortion”—Emphasizes that the patient’s suicide attempts were intended to influence someone else’s behavior or attitude; “Conflict with parents”—Contains a mention of problems with parents; “Conflict with (ex-)partner”—Contains references to problems with (ex-) partner; “Financial problems”—Contains links to financial and/or existential problems; “Social problems”—Contains references to the insufficient social and institutional background of the patient (e.g., homelessness); “Loneliness”—Emphasizes the lack of a close person, respectively, a sense of loneliness; “Distress”—Contains an emphasis on the fact that the patient “was no longer in control”.

The frequencies of individual themes in justification of reasons for suicidal behavior (Table 3) were generally different for the suicide attempters and the NSSIB patients. Pearson’s chi-square test revealed statistically significant differences between patients with NSSIB and those with SA for themes of non-communicated reasons (χ^2^ = 19.216; *p* < 0.001), loneliness (χ^2^ = 13.744; *p* < 0.001), social problems (χ^2^ = 8.04; *p* < 0.01), extortion (χ^2^ = 28.851; *p* < 0.001), and distress (χ^2^ = 14.466; *p* < 0.001). The results are shown in Table 3. The patients with SA most frequently did not communicate for any reason. The following other reasons in this group of patients were noticed: loneliness; distress; conflict (with partner and/or parents). Patients who engaged in NSSIB rarely refused to communicate the reasons and motives for suicidal behavior. The most common topics in NSSIB cases were extortion of others, conflicts (with partners and/or parents), and social problems. When investigating sex differences, the most frequent themes (including the rejection of specifying reasons) in suicide attempters and NSSIB cases did not differ significantly between men and women.

In the NSSIB group, there were 27 (out of 41) self-poisoning cases. In the SA patients’ group, there were 44 patients (out of 80) attempting suicide by self-poisoning. The χ^2^ independence test shows that the distribution of self-poisoning and the other types of self-harm in SA and NSSIB patients were not significantly different, showing no association between these variables of (χ^2^ = 1.31, *p* = 0.259) (Table 3).

One year following completion of the study, there were seven patients who died by suicide. Six deaths out of the seven were the patients from the SA group, one of them was from the NSSIB group.

## 4. Discussion

The results of the present study demonstrate a strong association of the character of suicidal behavior with mental disorder diagnosis. Concerning socio-demographics, living alone and the absence of social support increases the likelihood of a suicide attempt. Using a qualitative data analysis of patients’ statements, we identified different reasons and motives for suicidal behavior between patients with SA and patients with NSSIB.

There is no doubt that mental disorders are among the strongest predictors of suicidal behavior, including both attempted and completed suicides [17,28]. It has been repeatedly shown that patients with affective disorders represent a risk group in terms of suicidality. In this study, we aimed to find out the character of suicidal behavior among mental disorder diagnoses. Our results clearly show that diagnosis of affective disorders (major depressive disorder and bipolar affective disorder) is most frequently associated with a suicide attempt. According to Bradvik [29], depressive disorder is strongly related to both suicidal ideation and attempt, but it lacks specificity as a predictor. According to Szanto et al. [30], male gender, higher income, current depression, and current and worst lifetime suicidal ideation severity, cognitive control deficits, and low levels of non-planning impulsivity predicted fatal and near-fatal suicidal behavior in late-life depression. Recently, Li et al. [31] suggested that previous suicidal behaviors, mental disorders (severe depression, anxiety, alcohol/substance disorders), and environmental factors (school and family context, social support, stressful life events, etc.) could contribute to suicidal behaviors in patients with major depressive disorder. The incidence of suicide attempts and accomplished suicides seems to be higher in the case of patients with major depressive disorder compared to those with bipolar affective disorder [32,33]. A recent meta-analysis showed that major depression has the highest pooled suicide rate, while bipolar disorder has the lowest suicide rate [34]. The mean prevalence of bipolar disorder among suicide victims was found to be markedly lower than the mean prevalence of major depressive disorder [35]. This difference may be at least due to a higher prevalence of major depressive disorder than bipolar disorder among the patients suffering from mental disorders. In our sample, schizophrenia was shown to be the second most frequent diagnosis associated with the risk of SA.

We confirmed that diagnosis of personality disorder significantly increases the rate of NSSIB. It should be noted, that the majority of studies published so far have been conducted on adolescents. To our knowledge, this is the first study investigating the rate of SA vs. NSSIB in adult patients with personality disorder, who were admitted to the hospital with ICD-10 diagnoses X60–X81. Personality disorder is associated with a wide range of psychopathology, including unstable mood, impulsive behaviors, as well as unstable interpersonal relationships. Personality disorders are estimated to be present in more than 30% of the individuals who die by suicide and about 40% of the individuals who make suicide attempts [36]. Most patients with a personality disorder, despite having suicidal thoughts for long periods of time and multiple suicide attempts, never kill themselves [37]. Next, we investigated the relationships between socio-demographic variables and suicidal behavior. Marital status accounts for the largest variation in suicidal behavior. The married have a consistently lower risk of suicide attempts compared to those who are single. On the other hand, being divorced or separated was associated with a higher SA, however, this result did not reach statistical significance. The differences between the single and divorced and the single and widowed were not significant. According to the recent study by Øien-Ødegaard et al. [38], higher suicide risk among the divorced and separated points to suicide risk being associated with the ceasing of social ties. Another socio-demographic variable revealed to be associated with suicidal behaviors was the lack of social support, which increased the probability of SA by up to four times. Social support from family has been shown to be a strong predictor of both suicidal ideation and suicide attempts [39].

There are sparse data to investigate the differences in motives for suicidal behavior between patients with SA and patients with NSSIB. In our sample, the suicide attempters and the NSSIB individuals reported different themes in the justification of reasons for suicidal behavior. The SA patients most frequently did not communicate any reason or were silent about them, for example, in those patients in a severe depressive episode. The most frequently communicated themes in SA were loneliness, distress, and conflicts with partner and/or parents. The most common topics for the NSSIB cases were extortion of others, conflicts with partners and/or parents, and social problems.

Diagnosis of a NSSI proposed in the DSM-5 is restricted to physical damage to the body, while self-poisoning is not considered as a NSSI, even if there was no suicidal intent. Chartrand et al. [40] compared the correlates and outcomes of non-suicidal adults who self-cut to non-suicidal adults who intentionally self-poison. They found that, for the most part, people who self-cut and those who intentionally self-poison were similar on sociodemographic and clinical correlates. They suggested that consideration should be given to broadening the classification of NSSI to include other methods of self-harm without suicidal intent [40], which is in consonance with our findings.

It is valuable that there are already defined categories in the ICD-11 (World Health Organization, 2017) that seem to be very useful for clinical practice. There are three different categories: (1) intentional self-harm person indent to die (in our research group SA); (2) intentional self-harm, the person did not intend to die (in our research, NSSIB); and (3) intentional self-harm, not known or not determined if a person intended to die. We suggest that the introduction of these diagnostic categories will improve statistical reporting on suicidal behavior in the clinical population and provide new insights for the diagnosis and treatment of patients with mental disorders.

The main limitation of the study is the representativeness of the sample. Only hospitalized patients were involved and some patients with self-harm could be released before possible assessment. Assessment according to the study protocol was postponed by up to five days and during that time there could be a considerable change in the patient’s attitude to his/her suicidal behavior, and so the interpretation of motivational factors presented at the time of self-harm. The comparison of the assessment at the time of acute admission and assessment after some interval of hospitalization could provide important findings.

## 5. Conclusions

In conclusion, the diagnosis of affective disorder represents a strong predictor of SA, while the diagnosis of personality disorder significantly increases the rate of NSSIB. The absence of social support increases the likelihood of SA. Marriage is a protective factor for suicide. The reasons for suicidal behavior are quietly different between the patients engaged in SA and in NSSIB.

## 6. Implications

Suicidal attempts are common and strong indicators for the emergency admission of patients. The result of this study is in accordance with common clinical experience and current trends for the necessity of more precise classification and management of patients with different kinds of self-harm behavior. If the codes within “X” of the ICD-10, without more precise specification, are used, it could have a misleading effect on the treatment plan for individual patients, and even for general health statistics and governmental programs for suicide prevention.

## Figures and Tables

**Table 1 ijerph-19-06283-t001:** Socio-demographic characteristics of the patients.

Characteristics	*n*	%
**Sex**		
Men	67	55.4
Women	54	44.6
**Average age (years ± SD)**		
Men	35.5 ± 13.7	
Women	44.0 ± 16.6	
Total	39.3 ± 15.6	
**Marital status**		
Single	63	52.1
Married	39	32.2
Divorced	11	9.1
Widowed	8	6.6
**Education**		
Elementary	41	33.9
Secondary without graduation	39	32.2
University degree	8	6.6
**Having children**		
Yes	60	49.6
No	61	50.4
**Mental disorder diagnosis (main group code of ICD-10 classification)**		
F0.x Organic, including symptomatic, mental disorders	6	5
F1.x Mental and behavioral disorders due to psychoactive substance use	17	14
F2.x Schizophrenia, schizotypal, delusional, and other non-mood psychotic disorders	16	13.2
F3.x Mood (affective) disorders	37	30.6
F4.x Anxiety, dissociative, stress-related, somatoform and other nonpsychotic mental disorders	22	18.2
F6.x Disorders of adult personality and behavior	10	8.3
F7.x Intellectual disabilities	4	3.3
F9.x Behavioral and emotional disorders with onset usually occurring in childhood and adolescence	9	7.4
**Suicides by day of week**		
Monday	22	18.2
Tuesday	23	19
Wednesday	21	17.4
Thursday	10	8.3
Friday	16	13.2
Saturday	16	13.2
Sunday	13	10.7
**History of mental disorders**		
Yes	71	58.7
No	50	41.3
**Childhood adversity**		
Yes	45	37.2
No	76	62.8
**Intentional self-harm (ICD-10)**		
Intentional self-poisoning (X60-F69)	71	58.7
Intentional self-harm by sharp object (X78)	35	28.9
Intentional self-harm by hanging, strangulation (X70), by jumping from a high place (X80), by jumping or lying before moving object (X81)	15	12.4

**Table 2 ijerph-19-06283-t002:** Different themes in the justification of motives for suicidal behavior between patients engaged in suicide attempt (SA) and in non-suicidal self-injurious behavior (NSSIB) as revealed by qualitative data analysis of patients’ statements.

Theme	*n* (SA)	%	*n* (NSSIB)	%	χ^2^ (*p*)
Financial problems	6	6.45	2	2.44	1.608 (0.205)
Conflict with (ex-)partner	12	12.90	11	13.41	0.010 (0.920)
Conflict with parents	9	9.68	13	15.85	1.512 (0.218)
Non-communicated	22	23.66	1	1.22	19.21 (0.001)
Loneliness	17	18.28	1	1.22	13.74 (0.001)
Impulsive	2	2.15	7	8.54	3.643 (0.056)
Social problems	2	2.15	11	13.41	8.040 (0.005)
Extortion	8	8.60	36	43.90	28.85 (0.001)
Distress	15	16.13	0	0.00	14.46 (0.001)
Sum	93	100.00	82	100.00	

SA—Suicide attempt; NSSIB—Non-suicidal self-injurious behavior; some patients gave more reasons.

**Table 3 ijerph-19-06283-t003:** Distribution of self-poisoning and other types of self-harm (sharp objects, hanging, strangulation, jumping) in the group of patients with suicide attempt (SA) and non-suicidal self-injurious behavior (NSSIB).

	Self-Poisoning	Otherself-Harm	Total
SA	44	36	80
NSSIB	27	14	41
Total	71	50	121

## Data Availability

Data available on request due to privacy/ethical restrictions.

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
