# Peer review of "Analysis of Motives and Factors Connected to Suicidal Behavior in Patients Hospitalized in a Psychiatric Department"

_ijerph, 2022, doi:10.3390/ijerph19106283_

Round 1
Reviewer 1 Report
Please, see attached file

Author Response
Please, see the attachment.

Reviewer 2 Report
Suicide behavior: An emerging public health challenge - analysis of motives of suicidal behavior in patients hospitalized in a psychiatric clinic
This is an important contribution with relevant implications to suicide prevention among psychiatric patients. Nevertheless, I believe the following changes would improve the overall quality of the article:
- Please provide definitions of suicide / suicidal behavior / suicide intention / suicide ideation / suicide attempt, and how these different definitions may results in limitations of previous research when trying to systematize the epidemiology of suicide behavior, including how it’s measured.
- Please provide more detailed information regarding the objectives of the research.
- How many participants were under 15? Who gave them permission to participate in this study?
- Please include a procedures sub-topic explaining how data were collected.
- Please include information regarding the Consolidated criteria for reporting qualitative research (COREQ)’s checklist.
- Please include examples of categories, items or questions asked.
- C-SSRS is a validated and reliable instrument for US but no validated version was used for Slovakian participants? If not, authors must present reliability data for this instrument.
- “We carried out a qualitative data analysis of subjects' statements for 81 investigating the reasons behind suicidal behavior in a group of patients with suicide at- 82 tempt (SA) and patients with Non-Suicidal Self-Injurious Behavior (NSSIB)” – I didn’t see any qualitative results nor any examples of statement from participants to elucidate. Please include them.
- Table 1 – please include SD for age variable.
- How was childhood adversity measured? Please clarify.
- Please include a limitations section.
- Please include an implications section, especially focusing on the prevention and intervention from a public mental health perspective.
Best wishes.
Author Response
Please, see the attachment.

Round 2
Reviewer 1 Report
I would like to thank the authors for the effort made to address my comments. They have answered to all my questions and concerns, and I think that the paper can now be accepted.
Reviewer 2 Report
Thank you for implementing the requested changes, they have improved the overall quality of the article.